# Role of long intergenic non-protein coding RNA 01857 in hepatocellular carcinoma malignancy via the regulation of the microRNA-197-3p/anterior GRadient 2 axis

**Jiangang Bi**[1☯], **Yusheng Guo**[1☯], **Qi Li**[1], **Liping Liu**[1], **Shiyun Bao**[1], **Ping Xu**[2]*

**1** Department of Hepatopancreatobiliary Surgery, Shenzhen People's Hospital, The Second Clinical Medical College of Jinan University, The First Affiliated Hospital of Southern University of Science and Technology, Shenzhen, China, **2** Department of Endocrinology, Shenzhen People's Hospital, The Second Clinical Medical College of Jinan University, The First Affiliated Hospital of Southern University of Science and Technology, Shenzhen, China

☯ These authors contributed equally to this work.
* drxuping244@163.com

**Data Availability Statement:** All relevant data are within the manuscript and its Supporting Information files.

## Abstract

### Objective

This study investigates the differential expression and the mechanism of long intergenic non-protein coding RNA (LINC) 01857 in hepatocellular carcinoma (HCC) proliferation and apoptosis.

### Methods

LINC01857 expression in HCC tissues and cells was evaluated. In addition, gain-of and loss-of functions were carried out to assess HCC cell proliferation and apoptosis. After that, LINC01857 subcellular localization was predicted and verified. Additionally, the binding relations between LINC01857 and microRNA (miRNA)-197-3p and between miR-197-3p and anterior GRadient 2 (AGR2) were detected and confirmed. Besides, HCC cell proliferation and apoptosis were assessed after silencing LINC01857 or overexpressing AGR2. Next, levels of key factors in the AKT and ERK pathways were measured. Additionally, xenograft transplantation was also conducted to confirm the effect of LINC01857 in HCC.

### Results

LINC01857 was overexpressed in HCC. Silencing LINC01857 leads to a blockage in HCC cell proliferation but improved apoptosis. LINC01857 could competitively bind to miR-197-3p and thus upregulate AGR2. miR-197-3p was poorly expressed in HCC, while AGR2 was overexpressed. Mechanistically, downregulated miR-197-3p or overexpressed AGR2 were observed to attenuate the effect of the LINC01857 knockdown on suppressing cell proliferation and enhancing apoptosis. Moreover, LINC01857 activated the AKT and ERK pathways through the manipulation of the miR-197-3p/AGR2 axis in HCC.

**Funding:** This research was supported by funds from the Medical Science and Technology Research Foundation of Guangdong Province (A2020559). The funders had no role in study design, data collection and analysis, decision to publish, or preparation of the manuscript.

**Competing interests:** No competing financial interests exist.

## Conclusion

The results of this study indicated that LINC01857 was highly expressed in HCC, and it could improve HCC cell proliferation and reduce apoptosis via competitively binding to miR-197-3p, promoting AGR2 and upregulating the AKT and ERK pathways.

## Introduction

Hepatocellular carcinoma (HCC) refers to a prevalent and lethal carcinoma classified under chronic hepatic disorders that leads to an unfavorable prognostic aftermath and high mortality [1, 2]. As a complicated tumor with heterogeneity, HCC accounts for almost 90% of primary liver neoplasms has emerged as a challenging dilemma and health burden worldwide [3]. HCC is a unique cancer that typically arises from chronic liver disease, and its incidence rate depends on the complex interaction among the host, disease and environmental factors, and chronic hepatitis B or C virus infection is the main risk factor worldwide [4]. Among HCC individuals at an early stage, excision, hepatic transplantation, therapeutic drugs and ablation are possible treatments [5]. Unfortunately, regarding advanced therapies, curable therapies are inaccessible and might cost a large amount of expenditure but just exert little efficiency on prognosis and overall survival rate [6]. Against this backdrop, acquaintance with the potential crosstalk in HCC is of increasing importance.

Long non-coding RNAs (lncRNAs) mediate cellular activities and influence physiological development, thus participating in different diseases and even tumors [7]. Furthermore, lncRNAs are dysregulated in HCC and they affect HCC cell viability, invasion, vascularization and apoptosis, rendering them necessary choices in HCC detection and therapy [8]. LINC01857 overexpression results in induced lymph node metastasis, cancer cell viability, frustrating clinical outcomes and tumor growth and inhibits apoptosis in human cancers, including gastric cancer and pancreatic ductal adenocarcinoma [9, 10]. Furthermore, based on the effect of LINC01857, its downstream network has been studied.

Competing endogenous RNA (ceRNA) is of great significance in human tumors as it closely connects lncRNAs, microRNAs (miRNAs) and circular RNAs into a tight link [11]. For instance, LINC01857 acts as a tumor promotor in breast cancer by sponging miR-1281 to encourage cell biological behaviors [12], suggesting that LINC01857 could function as a ceRNA to competitively bind to a miR and become involved in cancers. Furthermore, miRs are recognized as HCC inhibitors or drivers since they have contributory or destructive effects on cancer cell viability, apoptosis, metastasis and the epithelial-to-mesenchymal transition (EMT) [13]. miR-197-3p is poorly expressed in HCC, indicating a repulsive prognosis, while overexpressed miR-197-3p could reduce HCC cell metastasis [14]. A recent study has noted that miR-197-3p forms a ceRNA network by connecting circular F-box and WD repeat domain containing 7 (FBXW7) and FBXW7-185aa protein to mediate triple-negative breast cancer progression [15], corroborating that miR-197-3p is a capable factor in ceRNA mechanism. Additionally, it is documented that as a downstream mRNA in HCC, anterior Gradient homolog 2 (AGR2) is exhausted by an lncRNA/miR interaction to protect against HCC mobility [16]. Thus far, no studies directly explored the relationship between LINC01857 and HCC, or the downstream mechanism; however, based on the above evidence, LINC01857 likely exacerbates HCC via the miR-197-3p/AGR2 axis. We hope that this experiment could provide novel insights into HCC treatment.

## Materials and methods

### Ethics statement

This study was performed with the approval of the Clinical Ethical Committee of Shenzhen People's Hospital. All procedures were strictly conducted in accordance with the code of ethics. The protocol was approved by the Institutional Animal Care and Use Committee of Shenzhen People's Hospital. Great efforts were exerted to reduce animal number used and their suffering. The written informed consent was obtained from each participant.

### Sample collection [16, 17]

From June 2017 to August 2019, 54 HCC patients undergoing surgery in Shenzhen People's Hospital were recruited in our experiment to collect the HCC and paracancerous tissues. The separated tissues were immediately deposited in -80˚C liquid nitrogen for further use. No patients received chemotherapy or radiotherapy before tissue excision. In addition, the clinical HCC staging was defined by the tumor node metastasis classification of malignant tumors by the Union for International Cancer Control.

### Cell culture [18]

Human normal hepatic epithelial cells (HL-7702 cells, American Type Culture Collection (ATCC), Manassas, Virginia, USA) and human HCC cell lines (Hep3B, SMMC7721, Huh7 and HepG2 cells, Cell Bank of Chinese Academy of Sciences, Shanghai, China) were all cultured in Dulbecco's modified Eagle's medium (Invitrogen Inc., Carlsbad, CA, USA) consisting of 10% fetal bovine serum (FBS, Invitrogen) in a 37˚C incubator with 5% $CO_2$.

### Cell treatment and grouping

Hep3B and HepG2 cells were injected with small hairpin-negative control (sh-NC) (sh-NC: 5′-AGAATAGCACGTGGCACACAC-3') and sh-LINC01857 lentiviral vectors (sh-LINC01857: 5′-GCCAAAGCCCTGGACATAAGA-3′) (Shanghai Jima Company, Shanghai, China) (the multiplicity of infection was 30) and then treated by puromycin (2 μg/mL, Sigma-Aldrich, Merck KGaA, Darmstadt, Germany) for 48 h to identify cells that were stably silenced. Reverse transcription quantitative polymerase chain reaction (RT-qPCR) was conducted to examine the transection efficiency. In addition, miR-197-3p mimic, miR-197-3p inhibitor (IN), IN-NC, pcDNA-NC and pcDNA-AGR2 (Shanghai GenePharma Co, Ltd, Shanghai, China) were all transfected into the cells following the instructions of Lipofectamine 3000 (Invitrogen). After 48 h, the cells were collected for the subsequent experiments.

### 3-(4, 5-dimethylthiazol-2-yl)-2, 5-diphenyltetrazolium bromide (MTT) assay [19]

Hep3B and HepG2 cell proliferation was analyzed using MTT assay. Cells ($5 \times 10^3$ cells/well) in different groups were seeded into 96-well plates, with 100 μL cells per well. After the transfection, the cells were stained with MTT staining solution, and then the medium was refreshed with dimethyl sulfoxide solution (150 μL). The optical density (OD) at 490 nm was examined using a microplate reader (Model 550, Bio-Rad, Inc., Shanghai, China). Each step was repeated at least 3 times.

## Colony formation assay [19, 20]

Hep3B and HepG2 cells (100 cells) in different groups were seeded into 6-well plates for 14 days, washed twice with phosphate buffer saline (PBS), fixed in 4% paraformaldehyde for 20 min, and incubated with 1% crystal violet solution for 30 min. Then, the cells were rinsed with running water, dried, photographed and analyzed with a camera (Nikon Instruments, Chiyoda-ku, Tokyo, Japan).

## Flow cytometry

Subsequently, 100 μL Hep3B and HepG2 cell solutions ($3 \times 10^5$ cells) were cultured with 5 μL fluorescein isothiocyanate-labeled Annexin V and 10 μL propidium iodide (Becton, Dickinson Company, NJ, USA) at room temperature in the dark for 15 min, and then mixed with 400 μL binding buffer solution for flow cytometry (Beckman Coulter, CA 92821, USA). The excitation wavelength was 488 nm, and $1 \times 10^4$ cells were counted.

## Enzyme-linked immunosorbent assay (ELISA)

The ELISA kits were applied to determine the levels of B-cell lymphoma-2 (Bcl-2, ab272102) and Bcl-2-associated X (Bax, ab199080) (both from Abcam Inc., Cambridge, MA, USA) in the cells following the provided instructions of the manufactures. Then, the levels of Bcl-2 and Bax in the samples were examined according to the optical density (OD) value. Simply put, all kit components were rewarmed to room temperature and 40 μL sample diluent was added to the enzyme-coated plate. Next, 100 μL enzyme-labeled reagent was added to each well. The membrane was sealed and incubated at 37˚C for 60 min. The liquid was then discarded, dried, and washed. Then 50 μL developer A was added to each well, and then 50 μL developer B, gently shaking. The color was developed in the dark at 37˚C for 15 min, and then 50 μL termination solution was added to each well. The OD value was detected using a microplate reader (Bio-Rad, Hercules, CA, USA).

## Fractionation of nuclear and cytoplasmic RNA assay [21]

According to the instructions of the NE-PER nuclear and cytoplasmic extraction reagent (Thermo, Scientific, Rockford, IL, USA), the nucleus and cytoplasm of the cells were separated. HepG2 and Hep3B cells ($10^6$) were resuspended in buffer C (20 mM Tris-GCl pH 7.5, 75 mM NaCl, 5 mM $MgCl_2$, 0.5% p/w sodium deoxycholate, 0.2% Triton, 1 mM dithiothreitol (DTT) and 0.5% glycerol), and a protease inhibitor cocktail (Sigma) and RNase inhibitor (Thermo Scientific) were added. Supernatants containing cytoplasmic lysates were harvested after centrifugation. Then, the sphere-forming nuclei were washed with PBS; while the granular nuclei were resuspended in buffer N (10 mM Tris-HCl pH 8, 25 mM NaCl, 5 mM $MgCl_2$, 1% p/w sodium deoxycholate, 1% Triton, 0.2% sodium dodecyl sulfate (SDS), 1 mM DTT) containing protease and RNase inhibitors and then sonicated.

## RNA fluorescence in situ hybridization (FISH)

LINC01857 subcellular localization was assessed using the FISH kits (BIS-P0001, Guangzhou BersinBio, Guangzhou, Guangdong China). A digoxigenin-labeled LINC01857 probe hybridization solution was added to the cell slides, and antagonistic LINC01857 probes were used as an NC. The cell slides were hybridized at 42˚C for 16 h, immersed in 2× sodium chloride-sodium citrate solution, soaked in 70% ethanol for 3 min and stained with 4', 6-diamidino-2-phenylindole for 10 min. The cell slides were photographed under a Zeiss LSM880 NLO confocal microscope (Leica, Solms, Germany).

## Dual-luciferase reporter gene assay

It was predicted that there were binding sites between LINC01857 and miR-197-3p and between miR-197-3p and AGR2 via the RNA22 database (https://cm.jefferson.edu/rna22/Interactive/) [22] and TargetSacn database (http://www.targetscan.org/vert_71/) [23]. Therefore, wild type (WT) and mutant type (MUT) vectors (Ambion, Austin, Texas, USA) of LINC01857 or AGR2 were respectively constructed. Next, the vectors were co-transfected with a miR-197-3p mimic or mimic NC into 293T cells (ATCC) via Lipofectamine 3000 (Invitrogen). After 48 h of transfection, cells were harvested using a dual-luciferase reporter gene assay system (Promega Corp., Madison, WI, USA) for the following detection of luciferase activity.

## RNA immunoprecipitation (RIP)

miR-197-3p mimic or miR-197-3p-NC was transfected into Hep3B or HepG2 cells by Lipofectamine 3000 (Invitrogen). The cell lysates were pre-sealed with protein G beads, incubated with anti-AGO G beads (Pierce, Waltham, MA, USA) for 90 min at 4˚C, and then centrifuged at 600 *g* for 1 min. Then, the beads were collected, washed five times with radio-immunoprecipitation assay (RIPA) buffer solution and resuspended in50 mmol/L Tris-HCl (pH 7.0). After 45 min of incubation at 70˚C, the beads were reversely cross-linked to extract RNA, which was quantified by RT-qPCR.

## RNA pull-down

miR-197-3p and NC were Biotinylated with biotinylated RNA mixture and T7/SP6 RNA polymerase. Biotinylated miR-197-3p and NC were transfected into Hep3B or HepG2 cells using Lipofectamine 3000 (Invitrogen). After 48 hours, the cells were collected and incubated with streptavidin-coupled beads. RT-qPCR was used to verify the expression of LINC01857 and AGR2 by RNeasy Mini Kit (cat# 74104, Qiagen).

## RT-qPCR

According to the manufacturers' instructions, the Trizol method (Invitrogen) was employed to separate the total RNA in tissue homogenates and cells. The RNA concentration was measured with a spectrophotometer (NanoVueTM, General Electric Company, Schenectady, NY, USA), and the RNA purity was evaluated by OD260/OD280 (1.8~2.0). Then 2 μg RNA was reverse transcribed into cDNA by RT kits (Invitrogen). Real-time PCR was carried out using an ABI7500 qPCR instrument (7500, ABI, Inc., Foster City, CA, USA). The reaction conditions included the cDNA template (1 μL), primers (1 μL), 2X SYBR Green Mix (10 μL, Shanghai GeneCore BioTechnologies Co., Ltd., Shanghai, China) and ddH$_2$O (8 μL). PCR was performed using an ABI 7500 system platform (Applied Biosystems, Inc., Carlsbad, CA, USA). The primer sequences (Table 1) were designed by Sangon Biotech Co., Ltd. (Shanghai, China). Glyceraldehyde-3-phosphate dehydrogenase (GAPDH) and U6 served as the internal reference and the 2$^{-\Delta\Delta CT}$ method was employed to measure the relative expression level.

## Western blot analysis

Cells were mixed with RIPA lysis buffer (Beyotime Biotechnology Co., Ltd, Shanghai, China) consisting of protease inhibitor cocktail (Sigma), lysed on ice 30 min and centrifuged at 13000 rpm for 10 min with the supernatant extracted. Then, the extracted proteins were quantified using a bicinchoninic acid kit (Beyotime). Subsequently, the same amount of proteins was transferred onto polyvinylidene fluoride membranes via an electrophoresis tank after SDS polyacrylamide gel electrophoresis. The membranes were removed, blocked with the correct

**Table 1. Primers sequence of RT-qPCR.**

| Gene | Sequence (5'-3') |
|------|------------------|
| LINC01857 | F: ATGGCACGATCTCGGCTCACTGCA |
| | R: TTATAGGTGTGAGCCCCAGCGCCCA |
| miR-197-3p | F: TTCACCACCTTCTCCACCCAGC |
| | R: GCTGGGTGGAGAAGGTGGTGAA |
| AGR2 | F: ATGGAGAAAATTCCAGTGTCAGCAT |
| | R: TTACAATTCAGTCTTCAGCAACT |
| GAPDH | F: ATGGTTTACATGTTCCAATATG |
| | R: TTACTCCTTGGAGGCCATGTGG |
| U6 | F: CGCTTCGGCAGCACATATAC |
| | R: AATATGGAACGCTTCACGA |

Note: RT-qPCR, reverse transcription quantitative polymerase chain reaction; LINC, long intergenic non-protein coding RNA; miR, microRNA; AGR2, anterior GRadient 2; GAPDH, glyceraldehyde-3-phosphate dehydrogenase; F, forward; R, reverse

amount of blocking reagent (PBST containing 3% FBS and 0.1% Tween-20) for 1 h and then incubated with the following primary antibodies (all from Abcam) at a 4°C shaker overnight: GAPDH (1: 2500, ab9485), p-protein kinase B (Akt) (1: 500, ab38449, 56kDa), p-extracellular signal-regulated kinase (ERK) (1: 1000, ab201015, 44kDa). Then, the membranes were cultivated with secondary antibody (ab182016, Abcam). Subsequently, the membranes were developed by Beyo ECL Plusd chemiluminescence and analyzed by Alpha Ease FC 4.0 (Alpha Innotech, CA, USA) to obtain the gray value of the bands.

## Xenografts tumors in nude mice [21]

Additionally, 24 female BALB/c nude mice (6 weeks old, 18–22 g) (Beijing Vital River Laboratory Animal Technology Co., Ltd., Beijing, China, SYXK (Beijing) 2017–0033) were housed in microisolation cages. The mice were divided into the sh-LINC01857 group (mice were subcutaneously injected with 0.2 mL PBS containing $5×10^6$ HepG2 cells infected with sh-LINC01857 on the right side of axillary) and the sh-NC group (mice were subcutaneously injected with 0.2 mL PBS containing $5×10^6$ HepG2 cells infected with sh-NC on the right side of axillary) (n = 12). The mice were monitored daily, and the tumor volume was measured every 3 days. The tumor volume was assessed as V ($mm^3$) = length×width$^2$×0.5. Then, 15 days after the injection [21], the mice were euthanized; pentobarbital sodium ($≥100$ mg/kg) was intraperitoneally injected for the tumor separation and weighing. Tumors from every 6 mice were selected for immunohistochemistry and tumors from the other 6 mice were selected for RT-qPCR.

## Immunohistochemistry [16]

Tumor tissues were sliced into paraffin-embedded sections (5 μm), dewaxed, dehydrated, subjected to antigen retrieval, and sealed with sheep serum (Beyotime) for 20 min with the sheep serum discarded. Subsequently, the sections were incubated with Ki67 primary antibody (ab16667, Abcam) at 4°C overnight. After that, the sections were incubated with Ki67 secondary antibody (ab205718, Abcam) for 1 h and then detected by DAB system (Beyotime).

## Statistical analysis

SPSS 21.0 (IBM Corp. Armonk, NY, USA) was utilized to analyze data. The results were expressed as mean ± standard deviation. The data were normally distributed. A t-test was

applied to perform comparisons between two groups, one-way or two-way analysis of variance (ANOVA) was used to compare different groups, and Sidak's multiple comparisons test or Tukey's post-hoc multiple comparisons test or Dunnett's multiple comparisons test was used to analyze pairwise comparisons after ANOVA. A difference was considered statistically significant at a p value $< 0.05$.

## Results

### LINC01857 is overexpressed in HCC

To determine the role of LINC01857 in HCC, the Starbase database (http://starbase.sysu.edu.cn/panCancer.php) predicted that LINC01857 was robustly expressed in HCC (Fig 1A). Therefore, LINC01857 expression was measured in HCC and paracancerous tissues obtained from Shenzhen People's Hospital, and the results revealed that LINC01857 was overexpressed in HCC tissues; moreover, according to the detection of LINC01857 expression in HCC cells and HL-7702 cells, LINC01857 was sufficiently upregulated in HCC cells as compared with that in HL-7702 cells (all $p<0.01$) (Fig 1B and 1C). In summary, LINC01857 was overexpressed in HCC and it might be associated with HCC tumorigenesis.

### LINC01857 silencing leads to a blockage in HCC cell proliferation and promotes apoptosis

To determine the role of LINC01857 in HCC, LINC01857 expression was successfully silenced in HCC cells via the administration of sh-LINC01857 (all $p<0.01$) (Fig 2A). Subsequently, MTT and colony formation assay indicated that HCC cell proliferation was reduced (all $p<0.01$) (Fig 2B and 2C); while flow cytometry revealed that apoptosis was promoted (all $p<0.01$) (Fig 2D). Moreover, the levels of pro-apoptotic protein Bax and the anti-apoptotic protein Bcl-2 were measured through ELISA, which revealed that the Bax level was upregulated while the Bcl-2 level was decreased (all $p<0.01$) (Fig 2E). These findings suggested that silencing of LINC01857 suppressed HCC cell proliferation but enhanced apoptosis.

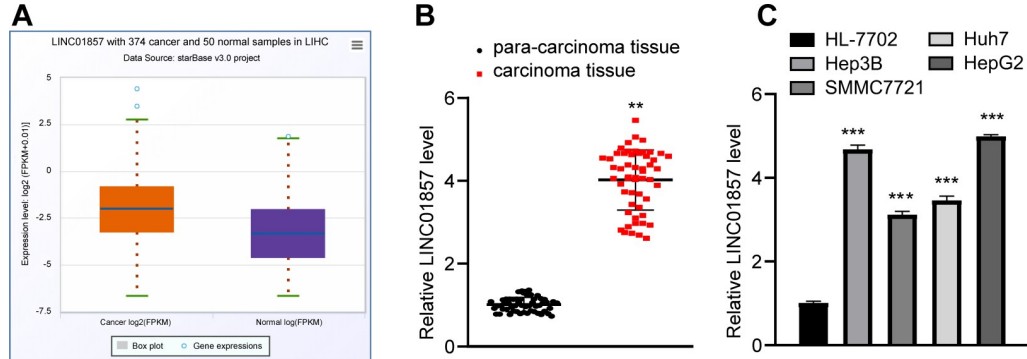

**Fig 1. LINC01857 is overexpressed in HCC.** A, LINC01857 expression in HCC predicted by the Starbase database. B, LINC01857 expression in HCC and paracancerous tissues detected via RT-qPCR, n = 54. C, LINC01857 expression in HCC cells examined by RT-qPCR. Three independent repeated cell tests were performed. The results were presented as mean ± standard deviation. The paired t-test was applied to analyze the data in panel B. One-way ANOVA was employed to analyze the data in panel C. Tukey's multiple comparisons test was employed for the post hoc test. ** $p<0.01$, *** $p<0.001$.

## LINC01857 competitively binds to miR-197-3p to improve AGR2 expression

To clarify the downstream molecular mechanism of HCC cell proliferation and apoptosis, the fractionation of nuclear and cytoplasmic RNA assay and FISH assay were performed, which verified that LINC01857 was mainly located at the cytoplasm (all $p<0.01$) (Fig 3A and 3B), illustrating that LINC01857 could influence HCC by regulating downstream genes through the ceRNA network. Thus, downstream miRNAs and mRNAs of LINC01857 were probed, and we noticed that there were binding relationships between LINC01857 and miR-197-3p and between miR-197-3p and AGR2. Low expression of miR-197-3p was observed in HCC [14]. The downstream target gene AGR2 of miR-197-3p was highly expressed in HCC [16, 21]. We hypothesized that LINC01857 might affect HCC by sponging miR-197-3p to modulate AGR2 expression. Firstly, RIP assay confirmed that LINC01857 could competitively bind to miR-197-3p (all $p < 0.01$, Fig 3C). Then, based on the binding sites between LINC01857 and miR-197-3p and between miR-197-3p and AGR2 (Fig 3D), a dual-luciferase reporter gene assay was carried out, which revealed that LINC01857 bound to miR-197-3p and miR-197-3p bound to AGR2 (all $p<0.01$) (Fig 3E). In addition, the binding relationships between LINC01857 and miR-197-3p, miR-197-3p and AGR2 were also verified by RNA pull-down assay (all $p < 0.01$, Fig 3F). Besides, our experiment revealed that miR-197-3p was weakly expressed in HCC; while AGR2 was overexpressed (all $p<0.01$) (Fig 3G). Besides, in HCC cells with LINC01857 silencing, miR-197-3p expression was upregulated and AGR2 was downregulated (all $p<0.01$) (Fig 3H). Therefore, LINC01857 could competitively bind to miR-197-3p to improve AGR2 expression in HCC.

## miR-197-3p knockdown attenuates the effect of silenced LINC01857 on reducing HCC cell proliferation and improving apoptosis

To confirm that LINC01857 could mediate HCC by sponging miR-197-3p, functional rescue assays were conducted as miR-197-3p expression was downregulated in HepG2 cells with silenced LINC01857 (all $p<0.01$) (Fig 4A) to evaluate cell proliferation and apoptosis. Then, we found that when miR-197-3p was downregulated, cell proliferation was enhanced (all $p<0.01$) (Fig 4B) while apoptosis was degraded (all $p<0.01$) (Fig 4C), indicating that downregulation of miR-197-3p reversed the effect of silencing LINC01857 on reducing cell proliferation and improving apoptosis in HCC.

## AGR2 overexpression attenuates the effect of silenced LINC01857 on reducing HCC cell proliferation and improving apoptosis

Similarly, to confirm the role of LINC01857 in HCC progression by controlling AGR2, AGR2 was overexpressed in HCC cells with silencing LINC01857 (all $p<0.01$) (Fig 5A), and HCC cell proliferation was potentiated (Fig 5B), but apoptosis was quenched (all $p<0.01$) (Fig 5C). These findings showed that AGR2 overexpression reversed the effect of silencing LINC01857 on reducing HCC cell proliferation and improving apoptosis.

## LINC01857 activates the AKT and ERK pathways by mediating the miR-197-3p/AGR2 axis

The investigations of the downstream mechanism of AGR2 revealed that AGR2 could upregulate the AKT and ERK pathways, which are related to the development of HCC [24, 25]. Therefore, levels of the key factors of the AKT and ERK pathways in HCC cells were assessed, and then it was discovered that the sh-LINC01857 group expressed downregulated phosphorylation levels of AKT and ERK in HCC cells (all $p<0.01$) (Fig 6A). However, in the sh-miR-197-3p-IN

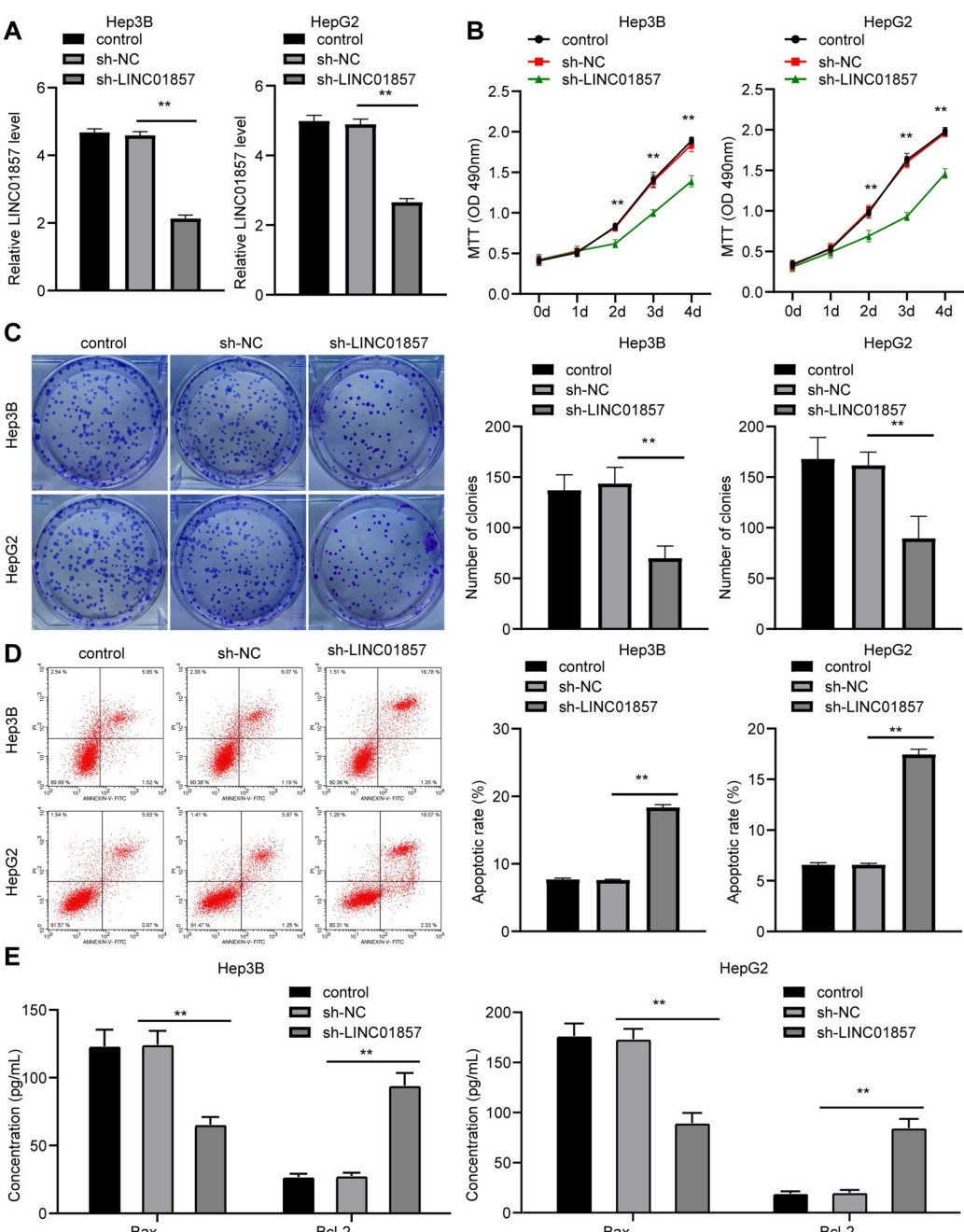

**Fig 2. LINC01857 inhibition inhibits HCC cell proliferation and promotes apoptosis.** HCC cells were infected with sh-LINC01857, with sh-NC as NC. A, LINC01857 expression in HCC cells verified by RT-qPCR. B and C, HCC cell proliferation detected by MTT (B) and colony formation assay (C). D, apoptotic rate assessed by flow cytometry. E, levels of Bax and Bcl-2 in HCC cells measured via ELISA. Three independent repeated tests were conducted. The results were presented as mean ± standard deviation. One-way ANOVA was employed to analyze the data in panels A, C and D. Two-way ANOVA was employed to analyze the data in panels B and E. Tukey's multiple comparisons test was applied for the post hoc test. ** $p < 0.01$.

and the sh-LINC01857+pc-AGR2 groups, the phosphorylation levels of AKT and ERK were evidently stimulated (all $p < 0.01$) (Fig 6B). In addition, the original unadjusted images of all imprints are presented in S1 Raw images. Overall, LINC01857 could activate the AKT and ERK pathways in HCC via the manipulation of the miR-197-3p/AGR2 axis.

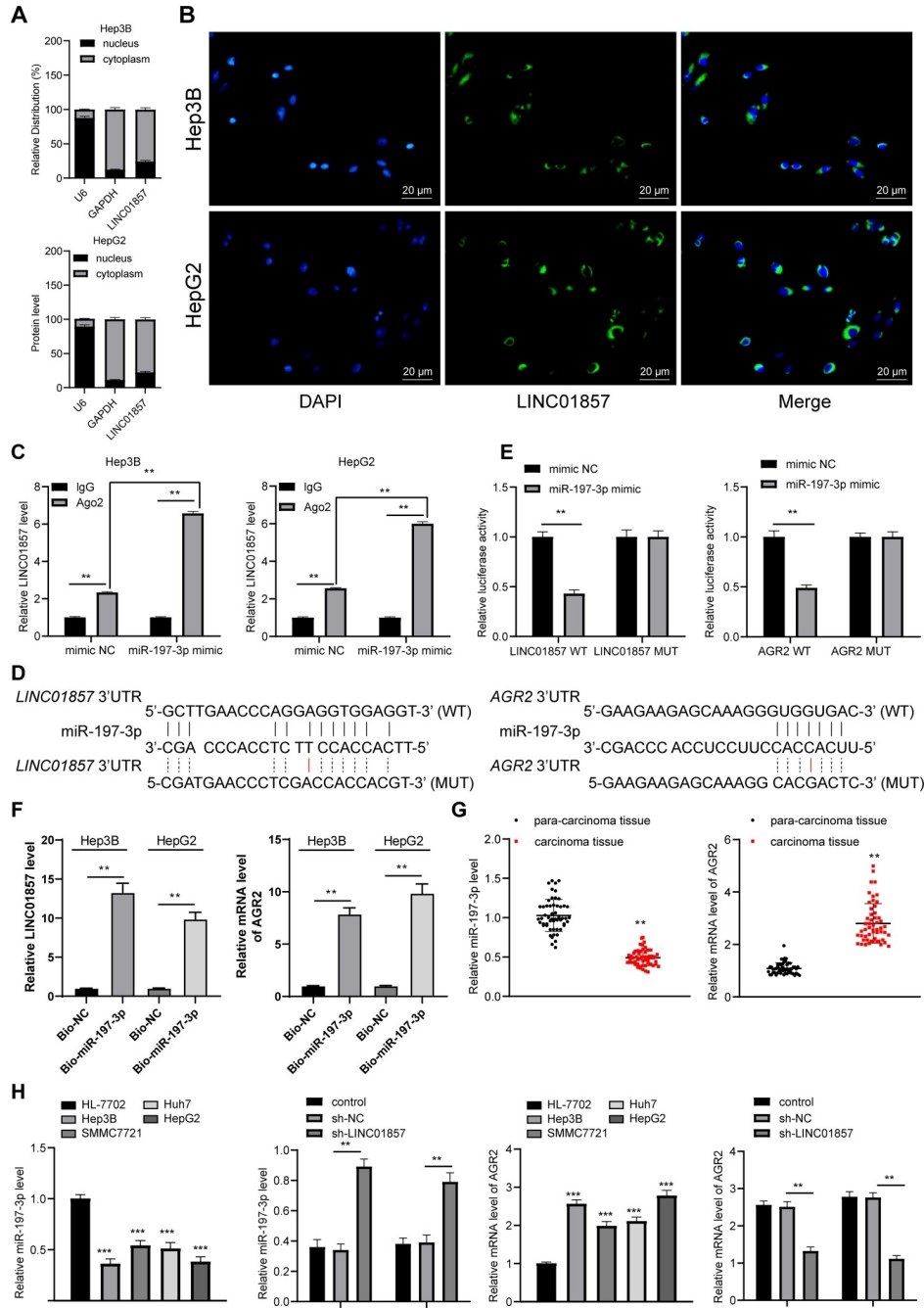

**Fig 3. LINC01857 competitively binds to miR-197-3p to improve AGR2 expression.** A, LINC01857 expression in HCC cells detected by fractionation of nuclear and cytoplasmic RNA and RT-qPCR. B, fluorescence localization of LINC01857 in HCC cells verified by RNA-FISH. C, binding relation between LINC01857 and miR-197-3p verified by RIP assay; D: binding sites between LINC01857 and miR-197-3p and between miR-197-3p and AGR2 predicted by RNA22 and TargetScan database. E-F, binding relations between LINC01857 and miR-197-3p and between miR-197-3p and AGR2 certified by RIP assay and dual-luciferase reporter gene assay. G, expression of miR-197-3p and AGR2 mRNA in HCC and paracancerous tissues examined by RT-qPCR, n = 54. H, expression of miR-197-3p and AGR2 mRNA in HCC cells assessed by RT-qPCR. Three independent repeated cell tests were conducted. The results were presented as mean ± standard deviation. The paired t-test was employed to analyze the data in panels G/F. One-way ANOVA was used to analyze the data in panel H. Tukey's multiple comparisons test was applied for post hoc test. Two-way ANOVA was employed to analyze the data in panels A, C, E, and H. Tukey's multiple comparisons test or Sidak's multiple comparisons test was employed for the post hoc test. ** $p<0.01$, *** $p<0.001$.

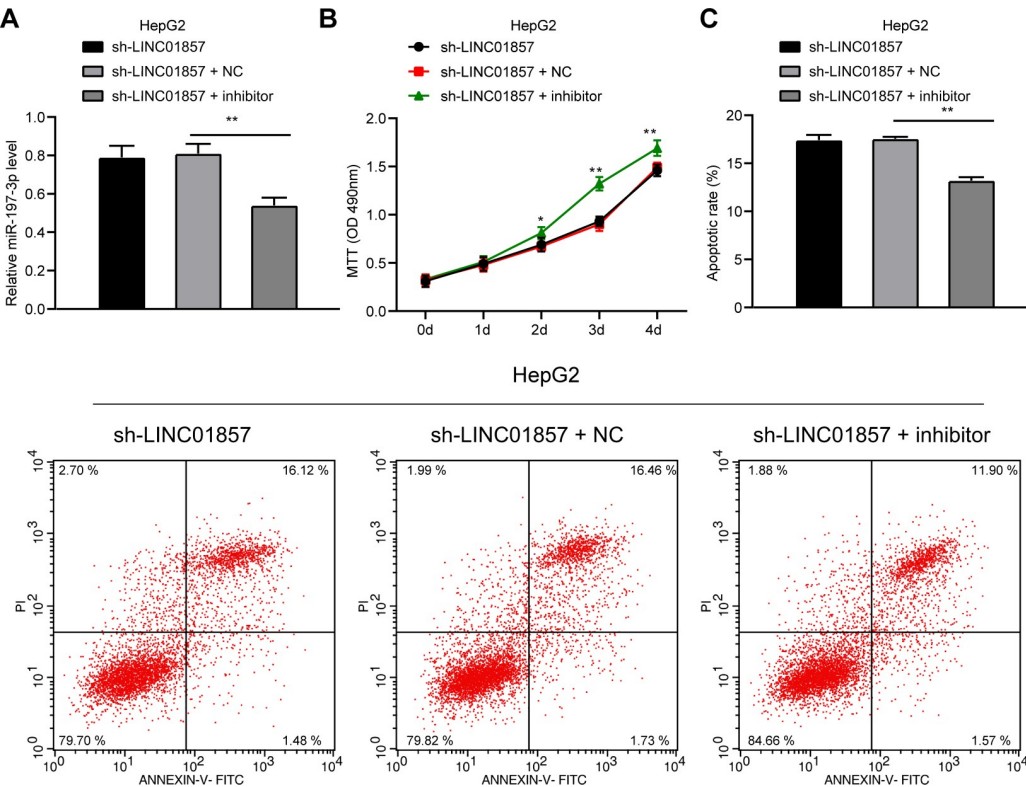

**Fig 4. miR-197-3p knockdown attenuates the effect of silenced LINC01857 on reducing HCC cell proliferation and improving apoptosis.** miR-197-3p-IN was transfected into HepG2 cells with silenced LINC01857, with IN-NC as NC. A, miR-197-3p inhibitor transfection efficiency detected by RT-qPCR. B, cell proliferation measured by MTT. C, apoptosis examined via flow cytometry. Three independent repeated tests were conducted. The results were presented as mean ± standard deviation. One-way ANOVA was employed to analyze the data in panel A. Two-way ANOVA was employed to analyze the data in panels B and C. Tukey's multiple comparisons test was applied for the post hoc test. ** $p < 0.01$.

## Silencing of LINC01857 inhibits HCC growth *in vivo*

Since the previous studies clarified that LINC01857 could affect HCC cell proliferation and apoptosis, we attempted to validate its role *in vitro* by establishing xenograft tumors in nude mice via the injection of HepG2 cells on the right side of axillary. Subsequently, our study revealed that compared with the sh-NC group, the sh-LINC01857 group exhibited repressed tumor growth (all $p < 0.01$) (Fig 7A) and reduced tumor weights ($p < 0.01$) (Fig 7B). The immunohistochemistry results showed that the sh-LINC01857 group possessed a decreased positive rate of Ki67 protein, which served as a marker of cell proliferation [26] ($p < 0.01$) (Fig 7C). Moreover, compared with the sh-NC group, LINC01857 was under-expressed, miR-197-3p was regulated and AGR2 mRNA was reduced in the sh-LINC01857 group (all $p < 0.01$) (Fig 7D–7F). The above results showed that silencing of LINC01857 could increase miR-197-3p expression to suppress AGR2 expression, thereby inhibiting HCC growth *in vivo*.

## Discussion

HCC is a malignant carcinoma that emerges in the interaction among disease, the environment and patients with chronic hepatic disorders, and it has caused the second largest number of deaths [4]. Studies have increasingly reported that lncRNAs are involved in a wide range of disorders, including neoplasms, via alterations in mRNA or miR expression, chromatin

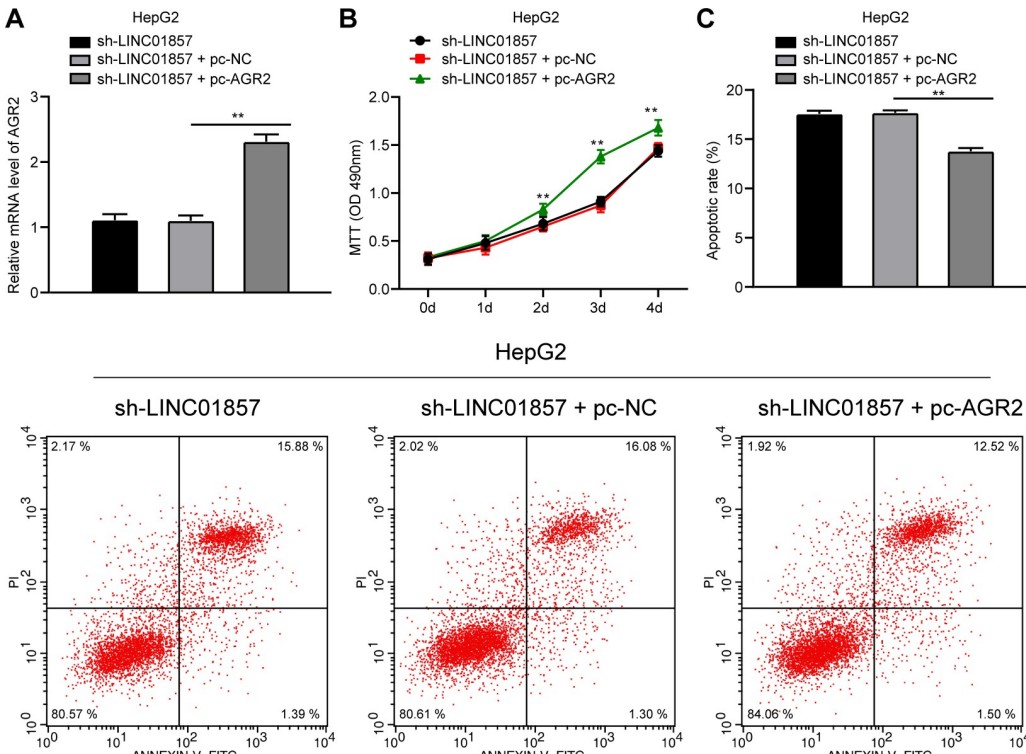

**Fig 5. AGR2 overexpression attenuates the effect of silenced LINC01857 on reducing HCC cell proliferation and improving apoptosis.** pcDNA-AGR2 was transfected into HepG2 cells with silenced LINC01857, with pcDNA-NC as NC. A, pcDNA-AGR2 transfection efficiency detected by RT-qPCR. B, cell proliferation measured by MTT. C, apoptosis examined via flow cytometry. Three independent repeated tests were conducted. The results were presented as mean ± standard deviation. One-way ANOVA was used to analyze the data in panel A. Two-way ANOVA was used to analyze the data in panels B and C. Tukey's multiple comparisons test was applied for the post hoc test. $**$ $p < 0.01$.

construction, gene scaffolding, gene transcription modulation and serving in the ceRNA network [27, 28]. As a member of lncRNAs, LINC01857 acts as a perilous signature in cancers ranging from breast cancer and glioma to diffuse large B-cell lymphoma as it augments cell mobility and viability [29]. To the best of our knowledge, the present study initially elucidated the expression pattern and mechanism of LINC01857 in LC cells.

The most principal finding of this study was that LINC01857 was strongly expressed in HCC. As it was recently reported, LINC01857 overexpression was observed in individuals with breast cancer, resulting in aggrandized cell expansion, aggressiveness, poor clinical aftermath and continuous apoptosis [30]. It was revealed that LINC01857 overexpression was correlated with the poor recurrence-free survival rate in HCC patients with fibrosis, illustrating that LINC01857 is detrimental to HCC development and prognosis [31]. Generally, LINC01857 is detrimental in some cancers. In this study, silencing LINC01857 expression limited HCC cell proliferation and promoted apoptosis as presented by the stimulated Bax level and reduced Bcl-2 level. Apoptosis represents a process of cell death aiming to balance the normal cells and pathological cells, and its absence might lead to neoplastic diseases or genomic disintegration [32]. Since apoptosis mainly affects injured, infected or redundant cells, it remains a conspicuous program that protects against hepatic oncogenesis [33]. Saleem and his colleagues noted that the activation of Bax and restriction of Bcl-2 were effective in inducing apoptosis to defend against cancers [34]. In diffuse large B-cell lymphoma, Li *et al*. noted that LINC01857 deteriorated tumor malignancy by reducing apoptosis [29]. Furthermore, when HCC cells were treated

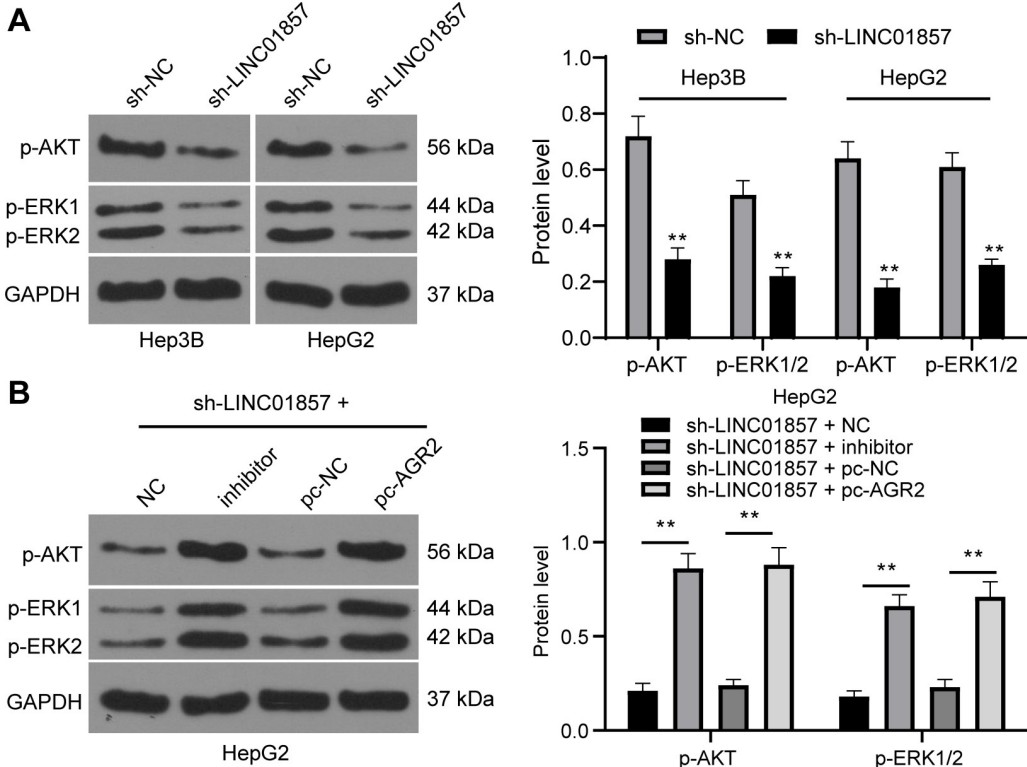

**Fig 6. LINC01857 activates the AKT and ERK pathways by mediating the miR-197-3p/AGR2 axis.** A and B, protein levels and phosphorylation levels of AKT and ERK in HCC cells from different groups verified by western blot analysis. Three independent repeated tests were conducted. The results were presented as mean ± standard deviation. Two-way ANOVA was employed to analyze the data. Tukey's multiple comparisons test was applied for the post hoc test. ** $p < 0.01$.

with anti-tumor drugs, the Bax level was improved while the Bcl-2 level was limited [35]. Therefore, silencing LINC01857 might effectively alleviate HCC progression.

LINC01857 could competitively bind to miR-197-3p to enhance AGR2 expression. It was reported that LINC01857 targets miR-200b to strengthen gastric cancer cell aggressiveness and dissemination [10]. Furthermore, Li *et al*. discovered that LINC01857 could competitively bind to miR-141-3p to upregulate mitogen-activated protein kinase 4, thereby accelerating the EMT in diffuse large B-cell lymphoma [29]. Besides, a recent study suggested that miR-197-3p is inactivated in colorectal cancer and functions as a sponge in a lncRNA/miR/mRNA crosstalk to sensitize cancer cells to curative drugs [36]. Similarly, miR-197-3p was specifically bound to tumor suppressor candidate 8 and EH-domain containing 2, thus to influence osteosarcoma growth and metastasis [37]. Our results showed that the expression of miR-197-3p is weak in LC tissues and cells. Previous studies have noted that AGR2 is expressed in many solid tumor types, including prostate, pancreatic, breast and lung cancer, and that the overexpression of AGR2 is closely associated with the growth, metastasis and survival of tumors (PMID: 27283903) [38, 39]. Collectively, the LINC01857/miR-197-3p/AGR2 axis in HCC cells was identified.

Moreover, weakly expressed miR-197-3p attenuated the effect of silenced LINC01857 on reducing HCC cell proliferation and improving apoptosis. According to the research developed by Cabral and his colleagues, miR-197-3p was decreased in HCC, which was accompanied by high metastasis and unwelcomed clinical results [40]. Besides, miR-197-3p was weakly expressed in primary biliary cirrhosis, suggesting that it was conducive to hepatic diseases

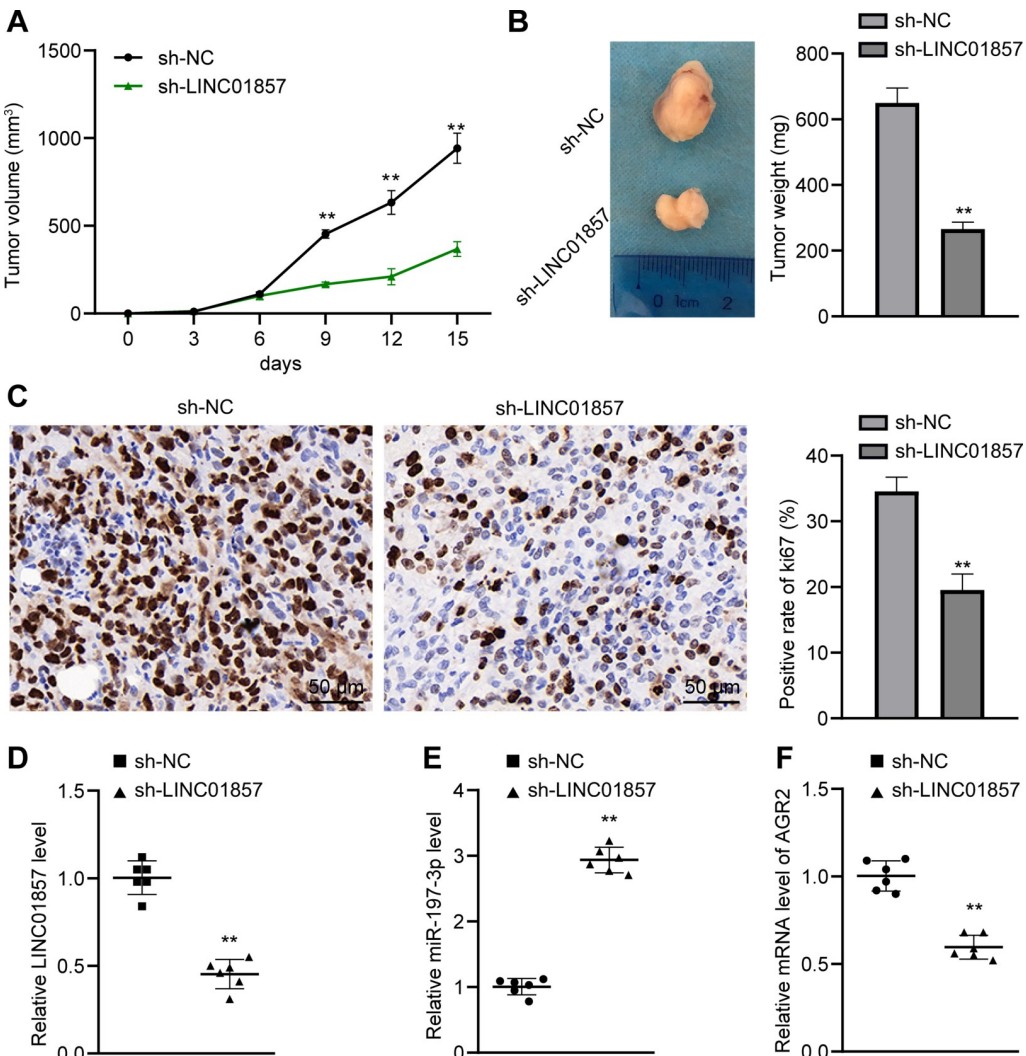

**Fig 7. Silencing of LINC01857 inhibits HCC growth *in vivo*.** A, tumor volume in HCC model, n = 12. B, representative images and weights of tumors, n = 6. C, ki67 positive rate of tumors in each group analyzed by immunohistochemistry, n = 6. D, E and F, expression of LINC01857, miR-197-3p and AGR2 mRNA detected by RT-qPCR, n = 6. The results in panels A, B and C were presented as mean ± standard deviation. The independent t-test was employed to analyze the data in panels B, C, D, E and F. Two-way ANOVA was employed to analyze the data in panel A. Tukey's multiple comparisons test was applied for the post hoc test. Compared with the si-NC group, ** $p < 0.01$.

[41]. When apoptosis was reduced in gastric cancer by overexpressed oncogenes, miR-197-3p was also decreased, illustrating the positive correlation between miR-197-3p and apoptosis [42]. Furthermore, AGR2 overexpression reversed the effect of silencing LINC01857 on reducing HCC cell proliferation and improving apoptosis. Notably, when AGR2 was upregulated by the competitive binding between LINC00460 and miR-342-3p, HCC expansion, aggressiveness and invasiveness were all enhanced [21]. AGR2 represses apoptosis in cancers, including pancreatic cancer, cervical cancer and breast cancer [43–45]. Significantly, LINC01857 activated the AKT and ERK pathways by mediating the miR-197-3p/AGR2 axis with the involvement of a high Ki67 protein positive rate. Lim *et al.* revealed that AKT and ERK knockdown brought about suppression of HCC vitality, aggressiveness and EMT [25]. The levels of AGR2, AKT, ERK and Ki67 were all decreased in pancreatic cancer with downregulated

tumor promotors, which confirmed the oncogenic role of the above cytokines [24]. Moreover, in endometrial cancer with overexpressed AGR2, Ki67-positive rate was accordingly elevated, indicating the detrimental role of Ki67 in tumors [46]. Thus, the modulation of the miR-197-3p/AGR2 axis and the AKT and ERK pathways by LINC01857 is necessarily involved in HCC growth.

## Conclusion

In summary, it was originally unearthed through this experiment that silencing of LINC01857 could alleviate HCC malignancy by participating in the ceRNA interaction to upregulate miR-197-3p and inactivate AGR2, thereby downregulating the AKT and ERK pathways. These findings unveiled a novel prevention strategy for HCC. However, there are still some limitations in the research, for example, We found that LINC01857 was highly expressed in LC tissues, and other studies could investigate the function of LINC01857 in future research. Other miRNAs downstream of LINC01857 and other target genes of miR-197-3p also need to be explored. In addition, the function of the AKT and ERK pathways isn't fully identified. Besides, this study represents a preclinical effort, and although our experiment offers therapeutic clues regarding HCC, the experimental findings and practical application in reality need further corroboration.

## Supporting information

**S1 Raw images. The phosphorylation levels of AKT and ERK in HCC cells were decreased after LINC01857 expression was silenced, while the phosphorylation levels of AKT and ERK were increased significantly after miR-197-3p downregulation or AGR2 overexpression in HCC cells with silencing LINC01857 expression.**
(PDF)

## Author Contributions

**Conceptualization:** Jiangang Bi.

**Data curation:** Yusheng Guo.

**Investigation:** Qi Li.

**Methodology:** Jiangang Bi.

**Software:** Shiyun Bao.

**Supervision:** Liping Liu.

**Validation:** Shiyun Bao.

**Visualization:** Qi Li.

**Writing – original draft:** Yusheng Guo.

**Writing – review & editing:** Ping Xu.

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
