## [Decision Letter · Decision Letter 0]

14 Jul 2021

PONE-D-21-13013

Role of long intergenic non-protein coding RNA 01857 in hepatocellular carcinoma malignancy via regulating the microRNA-197-3p/anterior GRadient 2 axis

PLOS ONE

Dear Dr. Xu,

Thank you for submitting your manuscript to PLOS ONE. After careful consideration, we feel that it has merit but does not fully meet PLOS ONE’s publication criteria as it currently stands. Therefore, we invite you to submit a revised version of the manuscript that addresses the points raised during the review process.

Please fully address both reviewers comments and concerns.

We look forward to receiving your revised manuscript.

Kind regards,

Olorunseun Ogunwobi, MD, PhD

Academic Editor

PLOS ONE

Journal Requirements:

2. Please provide additional details regarding participant consent. In the ethics statement in the Methods and online submission information, please ensure that you have specified (1) whether consent was informed and (2) what type you obtained (for instance, written or verbal, and if verbal, how it was documented and witnessed). If your study included minors, state whether you obtained consent from parents or guardians. If the need for consent was waived, please ensure that you have discussed whether all data were fully anonymized before you accessed them and/or whether the IRB or ethics committee waived the requirement for informed consent.

3. Please ensure you have thoroughly discussed any potential limitations of this study within the Discussion section, including the potential impact of confounding factors.

This research was supported by funds from the Medical Science and Technology Research Foundation of Guangdong Province (A2020559).

Reviewers' comments:

Reviewer's Responses to Questions

**Comments to the Author**

1. Is the manuscript technically sound, and do the data support the conclusions?

Reviewer #1: Yes

Reviewer #2: Yes

2. Has the statistical analysis been performed appropriately and rigorously? 

Reviewer #1: Yes

Reviewer #2: Yes

3. Have the authors made all data underlying the findings in their manuscript fully available?

Reviewer #1: Yes

Reviewer #2: Yes

4. Is the manuscript presented in an intelligible fashion and written in standard English?

Reviewer #1: Yes

Reviewer #2: No

5. Review Comments to the Author

Reviewer #1: The authors have shown that role of long intergenic non-protein coding RNA 01857 in hepatocellular carcinoma via regulating the microRNA-197-3p/AGR2. The work is really interesting, however there are a few minor mistakes which needs to be rectified.

1. Some of the fonts used in the figures are not clear (For eg, Fig 1a, 3a, 3d etc.). I recommend you to provide a better figure for 3c.

2. Please provide the sequences of siRNAs used in the study.

3. Mention the details of the ELISA kit used in this study.

4. How much total RNA was used for the reverse transcription?

5. It is better to write “miRNA”, not “miRs”.

6. Some typos are there in the manuscript.

a. “TRIzol”

b. In some places you have written “tumour” and in some other places “tumor”. Please maintain a consistency.

c. Spelling mistake: “behavirs”

Reviewer #2: The study is interesting and useful in deciphering the contributors to HCC development and progression. However, there are a number of obstacles in achieving clarity in the manuscript preparation including numerous grammatical errors. In the Introduction: “Except for chronic hepatitis B or C virus infection, unbalanced lifestyle in rich regions and the lack of dietary aflatoxin uptake in underdeveloped economies are all boosters for HCC expansion” is not clearly stated. It would be better to state that “…lack of dietary aflatoxin uptake and elimination….”

The sentence: “Furthermore, based on the effect of LINC01857, its downstream network is researched” needs more refining. It seems an incomplete sentence.

In the Methods: “Dual-luciferase reporter gene assay”- the letter “e” appear as “c”s. These substitutions between the letters “c” and “e” occur throughout the manuscript including in the Discussion.

In the Results: The caption: “LINC01857 under-expression inhibits HCC cell proliferation and promotes apoptosis” implies that silencing of LINC0857 is a direct mechanism, which at this point in the experiment is premature and not accurate. It is better stated that” LINC01857 under-expression leads to a blockage in HCC cell proliferation and promotes apoptosis.”

Fig. 1A is not clear. In fact, a number of the figures are blurry including #a,3B, etc.

Also, the Discussion section needs more alignment with the experimental results obtained instead of regurgitating the contents of the Introduction. The logistical writing of the manuscript will add significantly to the experimental output of the study which is very important.

The evidence for physical protein-protein interaction between LINC01857 and microRNA (miR)-197-3p and between miR-197-3p and anterior GRadient 2 (AGR2) needs more convincing results - perhaps using a pull down assay since this study seems to show a novel interaction.

6. PLOS authors have the option to publish the peer review history of their article (what does this mean?). If published, this will include your full peer review and any attached files.

Reviewer #1: No

Reviewer #2: No

---

## [Author Response · Author response to Decision Letter 0]

10 Aug 2021

Dear Editor and Reviewers, 

We thank you very much for giving us an opportunity to make the revision, and thank you very much for your comments and opinions. The reviewer’s comments were highly insightful and enabled us to greatly improve the quality of our manuscript. We have revised the manuscript, according to the comments and suggestions of reviewers and editor, and responded point by point to the comments. 

With regard to the reviewers’ comments and suggestions, we’d like reply as follows

Reviewers' comments:

Reviewer's Responses to Questions

Comments to the Author

1. Is the manuscript technically sound, and do the data support the conclusions?

Reviewer #1: Yes

Reviewer #2: Yes

2. Has the statistical analysis been performed appropriately and rigorously?

Reviewer #1: Yes

Reviewer #2: Yes

3. Have the authors made all data underlying the findings in their manuscript fully available?

Reviewer #1: Yes

Reviewer #2: Yes

4. Is the manuscript presented in an intelligible fashion and written in standard English?

Reviewer #1: Yes

Reviewer #2: No

5.Review Comments to the Author

Reviewer #1: The authors have shown that role of long intergenic non-protein coding RNA 01857 in hepatocellular carcinoma via regulating the microRNA-197-3p/AGR2. The work is really interesting, however there are a few minor mistakes which needs to be rectified.

1. Some of the fonts used in the figures are not clear (For eg, Fig 1a, 3a, 3d etc.). I recommend you to provide a better figure for 3c.

Response: Thank you for your review. We have modified the fonts and provided higher quality pictures to make the pictures look clearer and easier to understand. Please review again.

2. Please provide the sequences of siRNAs used in the study. 

Response: Thank you for your suggestion. We have added the sequence to the corresponding method. Please review again.

3. Mention the details of the ELISA kit used in this study. 

Response: Thanks for your reminding. We have mentioned the details of the ELISA kits.

4. How much total RNA was used for the reverse transcription?

Response: Thank you for your question. 2 μg RNA was reverse transcribed into cDNA, which we have described in the method.

5. It is better to write “miRNA”, not “miRs”.

Response: Thanks for your comments. We have replaced “miRs” with “miRNA”.

6. Some typos are there in the manuscript.

a. “TRIzol”

b. In some places you have written “tumour” and in some other places “tumor”. Please maintain a consistency.

c. Spelling mistake: “behavirs”

Response: Thanks very much for your review and comments. We have revised these typos in the manuscript.

Reviewer #2: The study is interesting and useful in deciphering the contributors to HCC development and progression. However, there are a number of obstacles in achieving clarity in the manuscript preparation including numerous grammatical errors. In the Introduction: “Except for chronic hepatitis B or C virus infection, unbalanced lifestyle in rich regions and the lack of dietary aflatoxin uptake in underdeveloped economies are all boosters for HCC expansion” is not clearly stated. It would be better to state that “…lack of dietary aflatoxin uptake and elimination….”

The sentence: “Furthermore, based on the effect of LINC01857, its downstream network is researched” needs more refining. It seems an incomplete sentence.

Response: Thanks. We have refined this sentence. Please review again.

In the Methods: “Dual-luciferase reporter gene assay”- the letter “e” appear as “c”s. These substitutions between the letters “c” and “e” occur throughout the manuscript including in the Discussion.

Response: Thanks. We have corrected these errors.

In the Results: The caption: “LINC01857 under-expression inhibits HCC cell proliferation and promotes apoptosis” implies that silencing of LINC0857 is a direct mechanism, which at this point in the experiment is premature and not accurate. It is better stated that” LINC01857 under-expression leads to a blockage in HCC cell proliferation and promotes apoptosis.”

Response: Thank you very much. We have revised this caption accordingly.

Fig. 1A is not clear. In fact, a number of the figures are blurry including #a,3B, etc.

Response: Thank you for your review. We have checked all the pictures and provided higher quality pictures.

Also, the Discussion section needs more alignment with the experimental results obtained instead of regurgitating the contents of the Introduction. The logistical writing of the manuscript will add significantly to the experimental output of the study which is very important.

Response: Thank you for your professional advice. We revised the discussion part to make it more logical. Please review it again.

The evidence for physical protein-protein interaction between LINC01857 and microRNA (miR)-197-3p and between miR-197-3p and anterior GRadient 2 (AGR2) needs more convincing results - perhaps using a pull down assay since this study seems to show a novel interaction.

Response: Thank you for your review. According to your suggestion, we supplemented the RNA pull-down experiment to confirm the binding relationship between LINC01857 and miR-197-3p and miR-197-3p and AGR2 (Fig. 3F), and further verified the ceRNA network composed of linc01857/miR-197-3p/AGR2 in hepatocellular carcinoma. Thanks again.

If you have any queries, please do not hesitate to contact us. 

Once again, thank you very much for your time spent on this manuscript. 

Looking forward to hearing from you. 

Your sincerely,

Ping Xu

drxuping244@163.com

---

## [Decision Letter · Decision Letter 1]

24 Sep 2021

Role of long intergenic non-protein coding RNA 01857 in hepatocellular carcinoma malignancy via the regulation of the microRNA-197-3p/anterior GRadient 2 axis

PONE-D-21-13013R1

Dear Dr. Xu,

We’re pleased to inform you that your manuscript has been judged scientifically suitable for publication and will be formally accepted for publication once it meets all outstanding technical requirements.

Kind regards,

Olorunseun Ogunwobi, MD, PhD

Academic Editor

PLOS ONE

Reviewers' comments:

Reviewer's Responses to Questions

**Comments to the Author**

1. If the authors have adequately addressed your comments raised in a previous round of review and you feel that this manuscript is now acceptable for publication, you may indicate that here to bypass the “Comments to the Author” section, enter your conflict of interest statement in the “Confidential to Editor” section, and submit your "Accept" recommendation.

Reviewer #1: All comments have been addressed

Reviewer #2: All comments have been addressed

2. Is the manuscript technically sound, and do the data support the conclusions?

Reviewer #1: Yes

Reviewer #2: Yes

3. Has the statistical analysis been performed appropriately and rigorously? 

Reviewer #1: Yes

Reviewer #2: Yes

4. Have the authors made all data underlying the findings in their manuscript fully available?

Reviewer #1: Yes

Reviewer #2: Yes

5. Is the manuscript presented in an intelligible fashion and written in standard English?

Reviewer #1: Yes

Reviewer #2: Yes

6. Review Comments to the Author

Reviewer #1: The authors have addressed all the comments. The manuscript has been improved significantly and is now ready for publication.

Reviewer #2: The authors addressed my concerns including adding ny suggested experiments to validate their results.

7. PLOS authors have the option to publish the peer review history of their article (what does this mean?). If published, this will include your full peer review and any attached files.

Reviewer #1: No

Reviewer #2: **Yes: **Zaki Sherif

---

## [Editor Report · Acceptance letter]

29 Sep 2021

PONE-D-21-13013R1 

Role of long intergenic non-protein coding RNA 01857 in hepatocellular carcinoma malignancy via the regulation of the microRNA-197-3p/anterior GRadient 2 axis 

Dear Dr. Xu:

I'm pleased to inform you that your manuscript has been deemed suitable for publication in PLOS ONE. Congratulations! Your manuscript is now with our production department. 

Kind regards, 

on behalf of

Dr Olorunseun Ogunwobi 

Academic Editor

PLOS ONE